

# 1 Ensemble models from machine learning: an example of wave runup
# 2 and coastal dune erosion

Tomas Beuzen[1], Evan B. Goldstein[2], Kristen D. Splinter[1]
[1]Water Research Laboratory, School of Civil and Environmental Engineering, UNSW Sydney, NSW,
Australia
[2]Department of Geography, Environment, and Sustainability, University of North Carolina at
Greensboro, Greensboro, NC, USA

*Correspondence to*: Tomas Beuzen (t.beuzen@unsw.edu.au)

**Keywords:** Gaussian Process; Narrabeen-Collaroy; Lidar; swash; dune impact model



**Abstract**
After decades of study and significant data collection of time-varying swash on sandy beaches, there is
no single deterministic prediction scheme for wave runup that eliminates prediction error — even
bespoke, locally tuned predictors present scatter when compared to observations. Scatter in runup
prediction is meaningful and can be used to create probabilistic predictions of runup for a given wave
climate and beach slope. This contribution demonstrates this using a data-driven Gaussian process
predictor; a probabilistic machine learning technique. The runup predictor is developed using one year
of hourly wave runup data (8328 observations) collected by a fixed LIDAR at Narrabeen Beach,
Sydney, Australia. The Gaussian process predictor accurately predicts hourly wave runup elevation
when tested on unseen data with a root mean-squared-error of 0.18 m and bias of 0.02 m. The
uncertainty estimates output from the probabilistic GP predictor are then used practically in a
deterministic numerical model of coastal dune erosion, which relies on a parameterization of wave
runup, to generate ensemble predictions. When applied to a dataset of dune erosion caused by a storm
event that impacted Narrabeen Beach in 2011, the ensemble approach reproduced ~85% of the observed
variability in dune erosion along the 3.5 km beach and provided clear uncertainty estimates around
these predictions. This work demonstrates how data-driven methods can be used with traditional
deterministic models to develop ensemble predictions that provide more information and greater
forecasting skill when compared to a single model using a deterministic parameterization; an idea that
could be applied more generally to other numerical models of geomorphic systems.



## 1   Introduction

Wave runup is important for characterizing the vulnerability of beach and dune systems and coastal infrastructure to wave action. Wave runup is typically defined as the time-varying vertical elevation of wave action above ocean water levels and is a combination of wave swash and wave setup (Holman, 1986; Stockdon et al., 2006). Most parameterizations of wave runup use deterministic equations that output a single value for either the maximum runup elevation in a given time period, $R_{max}$, or the elevation exceeded by 2% of runup events in a given time period, $R_2$, based on a given set of input conditions. In the majority of runup formulae, these input conditions are easily obtainable parameters such as significant wave height, significant wave period, and beach slope (Atkinson et al., 2017; Holman, 1986; Hunt, 1959; Ruggiero et al., 2001; Stockdon et al., 2006). However, wave dispersion (Guza and Feddersen, 2012), wave spectrum (Van Oorschot and d'Angremond, 1969), nearshore morphology (Cohn and Ruggiero, 2016), bore-bore interaction (García-Medina et al., 2017), tidal stage (Guedes et al., 2013),  and a range of other possible processes have been shown to influence swash zone processes. Since typical wave runup parameterizations do not account for these more complex processes, there is often significant scatter in runup predictions when compared to observations (e.g., Atkinson et al., 2017; Stockdon et al., 2006). Even flexible machine learning approaches based on extensive runup datasets or consensus-style 'model of models' do not resolve prediction scatter in runup datasets (e.g., Atkinson et al., 2017; Passarella et al., 2018b; Power et al., 2018). This suggests that the development of a perfect deterministic parameterization of wave runup, especially with only reduced, easily obtainable inputs (i.e., wave height, wave period, and beach slope), is improbable.

The resulting inadequacies of a single deterministic parameterization of wave runup can cascade up through the scales to cause error in any larger model that uses a runup parameterization. It therefore makes sense to clearly incorporate prediction uncertainty into wave runup predictions. In disciplines such as hydrology and meteorology, with a more established tradition of forecasting, model uncertainty is often captured by using ensembles (e.g., Bauer et al., 2015; Cloke and Pappenberger, 2009). The benefits of ensemble modelling are typically superior skill and the explicit inclusion of uncertainty in predictions by outputting a range of possible model outcomes. Commonly used methods of generating



ensembles include combining different models (Limber et al., 2018) or perturbing model parameters,
initial conditions and/or input data (e.g., via Monte Carlo simulations (e.g., Callaghan et al., 2013)).

An alternative approach to quantify prediction uncertainty is to incorporate scatter about a mean
prediction into model parameterizations. For example, wave runup predictions at every time step could
be modelled with a deterministic parameterization plus a noise component that captures the scatter
about the deterministic prediction caused by unresolved processes. If parameterizations are stochastic,
or have a stochastic component, repeated model runs (given identical initial and forcing conditions)
produce different model outputs – an ensemble – that represents a range of possible values the process
could take. This is broadly analogous to the method of "stochastic parameterization" used in the
weather forecasting community for sub-grid scale processes and parameterizations (Berner et al., 2017).
In these applications, stochastic parameterization has been shown to produce better predictions than
traditional ensemble methods and is now routinely used by many operational weather forecasting
centers (Berner et al., 2017; Buchanan, 2018).

Stochastically varying a deterministic wave runup parameterization to form an ensemble still requires
defining the stochastic term — i.e., the stochastic element that should be added to the predicted runup at
each model time step. An alternative to specifying a predefined distribution or a noise term added to a
parameterization is to learn and parameterize the variability in wave runup from observational data
using machine learning techniques. Machine learning has had a wide range of applications in coastal
morphodynamics research (Goldstein et al., 2018) and has shown specific utility in understanding swash
processes (Passarella et al., 2018b; Power et al., 2018) as well as storm driven erosion (Beuzen et al.,
2018; den Heijer et al., 2012; Goldstein and Moore, 2016; Palmsten et al., 2014; Plant and Stockdon,
2012). While many machine learning algorithms and applications are often used to optimize
deterministic predictions, a Gaussian process is a probabilistic machine learning technique that directly
captures model uncertainty from data (Rasmussen and Williams, 2006). Recent work has specifically
used Gaussian processes to understand coastal processes such as large scale coastline erosion (Kupilik
et al., 2018).




The work presented here is focused on using a Gaussian process to build a data-driven probabilistic
predictor of wave runup that includes estimates of uncertainty. While quantifying uncertainty in runup
predictions from data is useful in itself, the benefit of this methodology is in explicitly including the
uncertainty with the runup predictor in a larger model that uses a runup parametrization, such as a
coastal dune erosion model. Dunes on sandy coastlines provide a natural barrier to storm erosion by
absorbing the impact of incident waves and storm surge and helping to prevent or delay flooding of
coastal hinterland and infrastructure (Mull and Ruggiero, 2014; Sallenger, 2000; Stockdon et al., 2007).
The accurate prediction of coastal dune erosion is therefore critical for characterizing the vulnerability
of dune and beach systems and coastal infrastructure to storm events. A variety of methods are available
for modelling dune erosion including: simple conceptual models relating hydrodynamic forcing,
antecedent morphology and dune response (Sallenger, 2000); empirical dune-impact models that relate
time-dependent dune erosion to the force of wave impact at the dune (Erikson et al., 2007; Larson et al.,
2004; Palmsten and Holman, 2012); data-driven machine learning models (Plant and Stockdon, 2012);
and more complex physics-based models (Roelvink et al., 2009). In this study, we focus on dune-impact
models, which are simple, commonly used models that typically rely on a parameterization of wave
runup to model time-dependent dune erosion. As inadequacies in the runup parameterization can
jeopardize the success of model results (Overbeck et al., 2017; Palmsten and Holman, 2012; Splinter et
al., 2018), it makes sense to use a runup predictor that includes prediction uncertainty.

The overall aim of this work is to demonstrate how probabilistic data-driven methods can be used with
deterministic models to develop ensemble predictions, an idea that could be applied more generally to
other numerical models of geomorphic systems. **Sect. 2** first describes the Gaussian process model
theory.  In **Sect. 3** the Gaussian process runup predictor is developed. In **Sect. 4** an example application
of the Gaussian process predictor of runup inside a morphodynamic model of coastal dune erosion to
build a 'hybrid' model (Goldstein and Coco, 2015; Krasnopolsky and Fox-Rabinovitz, 2006) that can
generate ensemble output is presented. A discussion of the results and technique is provided in **Sect. 5**
followed by conclusions in **Sect. 6**. The data and code used to develop the Gaussian Process runup



predictor in this manuscript are publicly available at
https://github.com/TomasBeuzen/BeuzenEtAl_GP_Paper.





## 2 Gaussian Processes

### 2.1 Gaussian Process Theory

Gaussian processes (GPs) are data-driven, non-parametric models. A brief introduction to GPs is given here; for a more detailed introduction the reader is referred to Rasmussen and Williams (2006). There are two main approaches to determine a function that best parameterizes a process over an input space: 1) select a class of functions to consider, e.g., polynomial functions, and best fit the functions to the data (a parametric approach); or, 2) consider all possible functions that could fit the data, and assign higher weight to functions that are more likely (a non-parametric approach) (Rasmussen and Williams, 2006). In the first approach it is necessary to decide on a class of functions to fit to the data – if all or parts of the data are not well modelled by the selected functions, then the predictions may be poor. In the second approach there is an infinite set of possible functions that could fit a data set (imagine the number of paths that could be drawn between two points on a graph). A GP addresses the problem of infinite possible functions by specifying a probability distribution over the space of possible functions that fit a given dataset. Based on this distribution, the GP quantifies what function most likely fits the underlying process generating the data and gives confidence intervals for this estimate. Additionally, random samples can also be drawn from the distribution to provide examples of what different functions that fit the dataset might look like.

A GP is defined as a collection of random variables, any finite set of which has a multivariate Gaussian distribution. The random variables in a GP represent the value of the underlying function that describes the data, $f(x)$, at location $x$. The typical workflow for a GP is to define a prior distribution over the space of possible functions that fit the data, form a posterior distribution by conditioning the prior on observed input/output data pairs ("training data"), and to then use this posterior distribution to predict unknown outputs at other input values ("testing data"). The key to GP modelling is the use of the multivariate Gaussian distribution, which has simple closed form solutions to the aforementioned conditioning process, as described below.




Whereas a univariate Gaussian distribution is defined by a mean and variance (i.e., $\mathcal{N}(\mu,\sigma^2)$), a GP (a
multivariate Gaussian distribution) is completely defined by a mean function *m(x)* and covariance
function *k(x, x')* (also known as a "kernel"), and is typically denoted:

$$f(\boldsymbol{x}) \sim \mathcal{N}(m(\boldsymbol{x}), k(\boldsymbol{x}, \boldsymbol{x}'))$$ (1)

Where $\boldsymbol{x}$ is an input vector of dimension $D$ ($\boldsymbol{x} \in \mathbb{R}^D$), and $f$ is the unknown function describing the data.
Note that for the remainder of this paper, a variable denoted in bold text represents a vector. The mean
function, *m(x)*, describes the expected mean value of the function describing the data at location $\boldsymbol{x}$,
while the covariance function encodes the correlation between the function values at locations in $\boldsymbol{x}$.

These concepts of GP development are further described using a hypothetical dataset of significant
wave height ($H_s$) versus wave runup ($R_2$) shown in **Fig. 1A**. The first step of GP modelling is to
constrain the infinite set of functions that could fit a dataset by defining a prior distribution over the
space of functions. This prior distribution encodes belief about what the underlying function is expected
to look like (e.g., smooth/erratic, cyclic/random, etc.) before constraining the model with any observed
training data. Typically it is assumed that the mean function of the GP prior, *m(x)*, is 0 everywhere, to
simplify notation and computation of the model (Rasmussen and Williams, 2006). Note that this does
not limit the GP posterior to be a constant mean process. The covariance function, *k(x,x')*, ultimately
encodes what the underlying functions look like because it controls how similar the function value at
one input point is to the function value at other input points.

There are many different types of covariance functions or "kernels". One of the most common, and the
one used in this study, is the squared exponential covariance function:

$$k\left(x_i, x_j\right) = \sigma_f^2 \exp\left[-\sum_{d=1}^{D} \frac{1}{2l_d^2}\left(x_{d,i} - x_{d,j}\right)^2\right]$$ (2)





Where $\sigma_f$ is the signal variance and $l$ is known as the length-scale, both of which are hyperparameters in
the model that can be estimated from data (discussed further in **Sect. 2.2**). Together the mean function
and covariance function specify a multivariate Gaussian distribution:

$f(\boldsymbol{x}) \sim \mathcal{N}(\mathbf{0}, K)$ (3)

Where $f$ is the output of the prior distribution, the mean function is assumed to be **0** and $K$ is the
covariance matrix made by evaluating the covariance function at arbitrary input points that lie within
the domain being modelled (i.e., $K(x,x)_{i,j} = k(x_i,x_j)$). Random sample functions can be drawn from this
prior distribution as demonstrated in **Fig. 1B.**

The goal is to determine which of these functions actually fit the observed data points (training data) in
**Fig. 1A**. This can be achieved by forming a posterior distribution on the function space by conditioning
the prior with the training data. Roughly speaking, this operation is mathematically equivalent to
drawing an infinite number of random functions from the multivariate Gaussian prior (**Eq. (3)**), and
then rejecting those that do not agree with the training data. As mentioned above, the multivariate
Gaussian offers a simple, closed form solution to this conditioning. Assuming that our observed training
data is noiseless (i.e., $y$ exactly represents the value of the underlying function $f$) then we can condition
the prior distribution with the training data samples $(\boldsymbol{x},\boldsymbol{y})$ to define a posterior distribution of the
function value $(\boldsymbol{f}_*)$ at arbitrary test inputs $(\boldsymbol{x}_*)$:

$\boldsymbol{f}^*|\boldsymbol{y} \sim \mathcal{N}(K_* K^{-1} y, K_{**} - K_* K^{-1} K_*^T)$ (4)

Where $\boldsymbol{f}_*$ is the output of the posterior distribution at the desired test points $\boldsymbol{x}_*$, $\boldsymbol{y}$ is the training data
outputs at inputs $\boldsymbol{x}$, $K_*$ is the covariance matrix made by evaluating the covariance function (**Eq. (2)**)
between the test inputs $\boldsymbol{x}_*$ and training inputs $\boldsymbol{x}$ (i.e., $k(\boldsymbol{x}_*,\boldsymbol{x})$), $K$ is the covariance matrix made by
evaluating the covariance function between training data points $\boldsymbol{x}$, and $K_{**}$ is the covariance matrix
made by evaluating the covariance function between test points $\boldsymbol{x}_*$. Function values can be sampled



from the posterior distribution as shown in **Fig. 1C**. These samples represent random realizations of
what the underlying function describing the training data could look like.

As stated earlier, in **Eq. (4)** and **Fig. 1C** there is an assumption that the training data is noiseless and
represents the exact value of the function at the specific point in input space. In reality, there is error
associated with observations of physical systems, such that:

$y = f(x) + \varepsilon$                                     (5)

Where $\varepsilon$ is assumed to be independent identically distributed Gaussian noise with variance $\sigma_n^2$. This
noise can be incorporated into the GP modelling framework through the use of a white noise kernel that
adds an element of Gaussian white noise into the model:

$k(x_i, x_j) = \sigma_n^2 \delta_{ij}$                                     (6)

Where $\sigma_n^2$ is the variance of the noise and $\delta_{ij}$ is a Kronecker delta which is 1 if $i = j$ and 0 otherwise.
The squared exponential kernel and white noise kernel are closed under addition and product
(Rasmussen and Williams, 2006), such that they can simply be combined to form a custom kernel for
use in the GP:

$k(x_i, x_j) = \sigma_f^2 \exp\left\{-\sum_{d=1}^{D} \frac{1}{2l_d^2}(x_{d,i} - x_{d,j})^2\right\} + \sigma_n^2 \delta_{ij}$                                     (7)

The combination of kernels to model different signals in a dataset (that vary over different spatial or
temporal timescales) is common in applications of GPs (Rasmussen and Williams, 2006; Reggente et
al., 2014; Roberts et al., 2013).  Samples drawn from the resultant "noisy" posterior distribution are
shown in **Fig. 1D** in which the GP can now be seen to not fit the observed training data precisely.



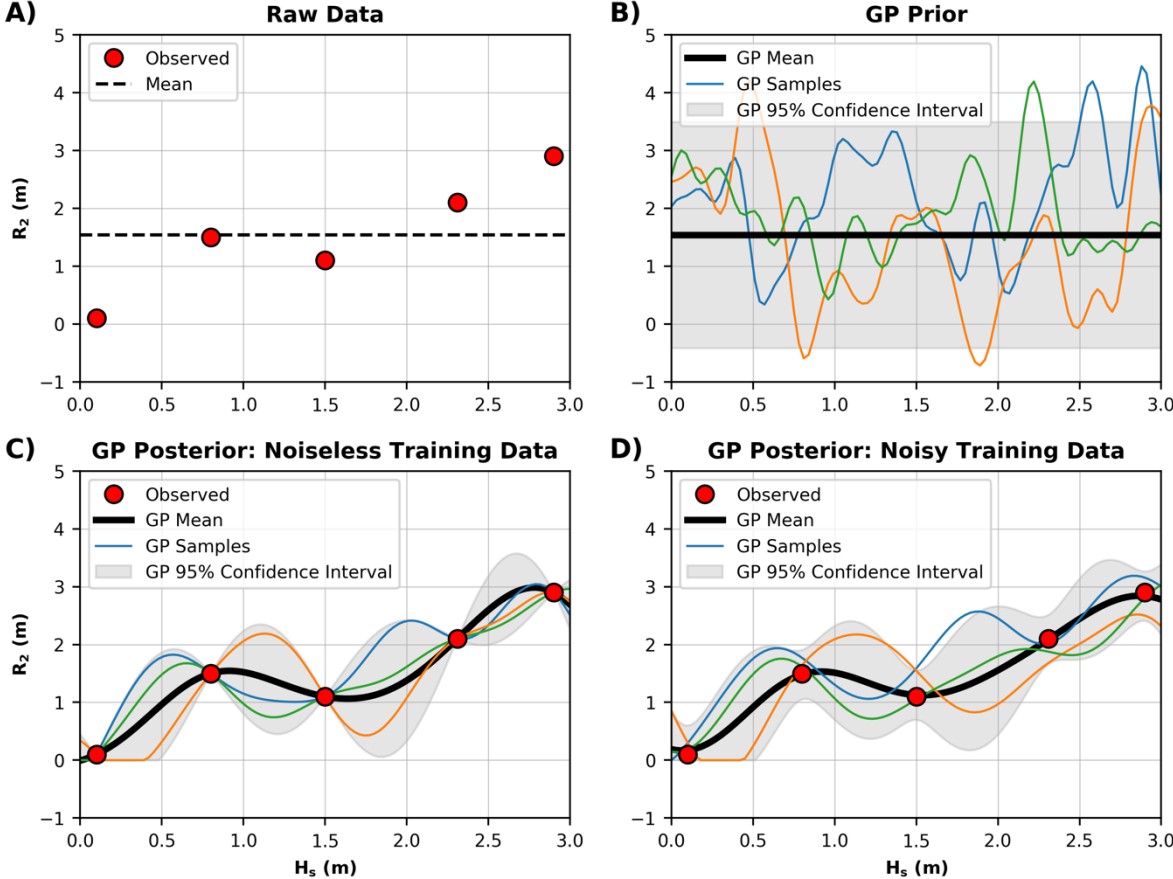

Fig. 1: A) Five hypothetical random observations of significant wave height ($H_s$) and 2% wave runup elevation ($R_2$). B) The Gaussian process (GP) prior distribution. C) The GP posterior distribution, formed by conditioning the prior distribution in (B) with the observed data points in (A), assuming the observations are noise-free. D). The GP posterior distribution conditioned on the observations with a noise component.

## 2.2 Gaussian Process Kernel Optimization

In **Eq. (7)** there are three hyperparameters: the signal variance ($\sigma_f$), the length scale ($l$) and the noise variance ($\sigma_n$). These hyperparameters are typically unknown but can be estimated and optimized based on the particular dataset. Here, this optimization is performed by using the typical methodology of maximizing the log-marginal-likelihood of the observed data $y$ given the hyperparameters:



$\log p(y|x, \sigma_f, l, \sigma_n)$                                                                          (8)

The Python toolkit SciKit-Learn (Pedregosa et al., 2011) was used to develop the GP described in this
study.

## 2.3   Training a Gaussian Process Model

It is standard practice in the development of data-driven machine learning models to divide the available
dataset into training, validation and testing subsets. The training data is used to fit model parameters.
The validation data is used to evaluate model performance and the model hyperparameters are usually
varied until performance on the validation data is optimized. Once the model is optimized, the
remaining test dataset is used to objectively evaluate its performance and generalizability. A decision
must be made about how to split a dataset into training, validation and testing subsets. There are many
different approaches to handle this splitting process; for example, random selection, cross-validation,
stratified sampling, or a number of other deterministic sampling techniques (Camus et al., 2011). The
exact technique used to generate the data subsets often depends on the problem at hand. Here, there
were two constraints to be considered; first, the computational expense of GPs scales by $O(n^3)$
(Rasmussen and Williams, 2006), so it is desirable to keep the training set as small as possible without
deteriorating model performance; and, secondly, machine learning models typically perform poorly
with out-of-sample predictions (i.e., extrapolation), so it is desirable to include in the training set the
data samples that captures the full range of variability in the data. Based on these constraints, we used a
maximum dissimilarity algorithm (MDA) to divide the available data into training, validation and
testing sets.

The MDA is a deterministic routine that iteratively adds a data point to the training set based on how
dissimilar it is to the data already included in the training set. Camus et al. (2011) provide a
comprehensive introduction to the MDA selection routine and it has been previously used in ML studies
(e.g., Goldstein et al., 2013). Briefly, to initialize the MDA routine, the data point with the maximum
sum of dissimilarity (defined by Euclidean distance) to all other data points is selected as the first data



point to be added to the training data set. Additional data points are included in the training set through
an iterative process whereby the next data point added is the one with maximum dissimilarity to those
already in the training set - this process continues until a user-defined training set size is reached. In this
way the MDA routine produces a set of training data that captures the range of variability present in the
full dataset. The data not selected for the training set are equally and randomly split to form the
validation dataset and test dataset.





## 3    Development of a Gaussian Process Runup Model

### 3.1    Runup Data

In 2014, an extended-range LIDAR (LIght Detection And Ranging) device (SICK LD-LRS 2110) was permanently installed on the rooftop of a beachside building (44 m above mean sea level) at Narrabeen-Collaroy Beach (hereafter referred to simply as Narrabeen) on the south-east coast of Australia (**Fig. 2**). Since 2014, this LIDAR has continuously scanned a single cross-shore profile transect extending from the base of the beachside building to a range of 130 m, capturing the surface of the beach profile and incident wave swash at a frequency of 5 Hz in both daylight and non-daylight hours. Specific details of the LIDAR setup and functioning can be found in (Phillips et al., 2019).

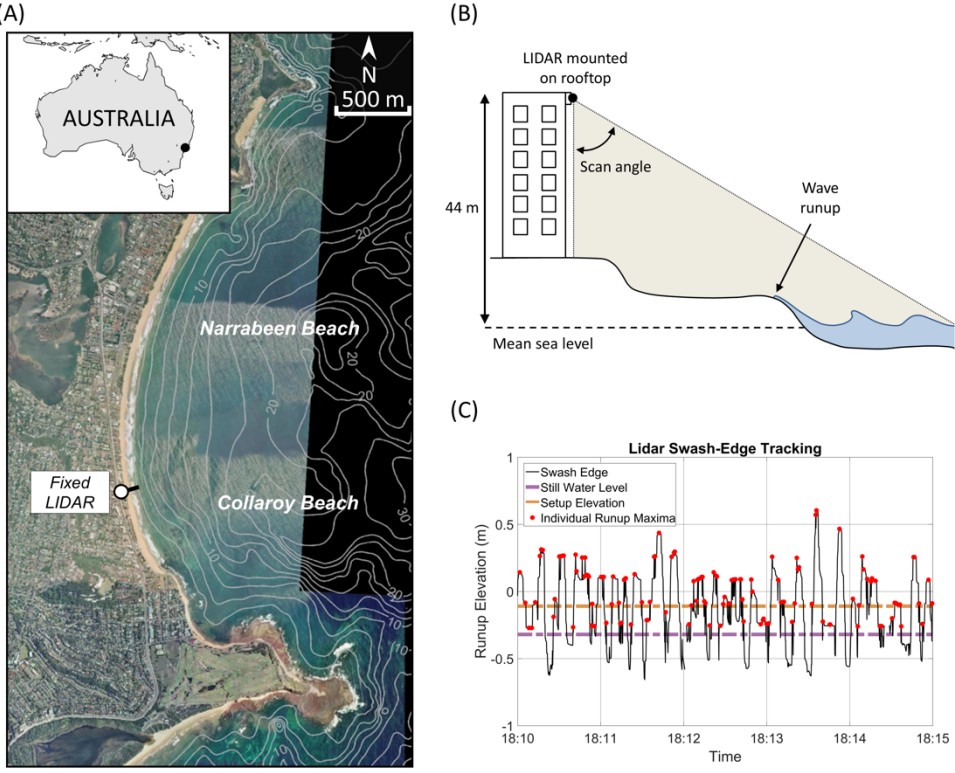

**Fig. 2: A) Narrabeen Beach, located on the southeast coast of Australia. B) Conceptual figure of the fixed LIDAR setup. C) A five-minute extract of runup elevation extracted from the LIDAR data, individual runup maxima are marked with red circles.**




Narrabeen Beach is a 3.6 km long embayed beach bounded by rocky headlands. It is composed of fine
to medium quartz sand (D50 ≈ 0.3 mm), with a ~30% carbonate fraction. Offshore, the coastline has a
steep and narrow (20 – 70 km) continental shelf (Short and Trenaman, 1992). The region is microtidal
and semidiurnal with a mean spring tidal range of 1.6 m and has a moderate to high energy deep water
wave climate characterized by persistent long-period SSE swell waves that is interrupted by storm
events (significant wave height > 3 m) typically 10 – 20 times per year (Short and Trenaman, 1992). In
the present study, approximately one year of the high-resolution wave runup LIDAR dataset available at
Narrabeen is used to develop a data-driven parameterization of the 2% exceedance of wave runup ($R_2$).
Data used to develop this parameterization were at hourly resolution and include: $R_2$, the beach slope
($\beta$), offshore significant wave height ($H_s$), and peak wave period ($T_p$). These data are described below
and have been commonly used to parameterize $R_2$ in other empirical models of wave runup (e.g.,
Holman, 1986; Hunt, 1959; Stockdon et al., 2006).

Individual wave runup elevation on the beach profile was extracted on a wave-by-wave basis from the
LIDAR dataset (**Fig. 2C**). Hourly $R_2$ was calculated as the 2% exceedance value for a given hour of
wave runup observations. $\beta$ was calculated as the linear (best-fit) slope of the beach profile over which
two standard deviations of wave runup values were observed during the hour. Hourly $H_s$ and $T_p$ data
were obtained from the Sydney Wave Rider buoy, situated 11 km offshore of Narrabeen in ~ 80 m
water depth. Narrabeen is an embayed beach, where prominent rocky headlands both attenuate and
refract incident waves. To remove these effects in the wave data and to emulate an open coastline and
generalize the parameterization of $R_2$ presented in this study, offshore wave data were first transformed
to a nearshore equivalent (10 m water depth) using the SWAN spectral wave model (Booij et al., 1999),
and then reverse shoaled back to deep water wave data. A total of 8328 hourly samples of $R_2$, $\beta$, $H_s$ and
$T_p$ were extracted to develop a parameterization of $R_2$ in this study. Histograms of this data are shown in
**Fig. 3**.



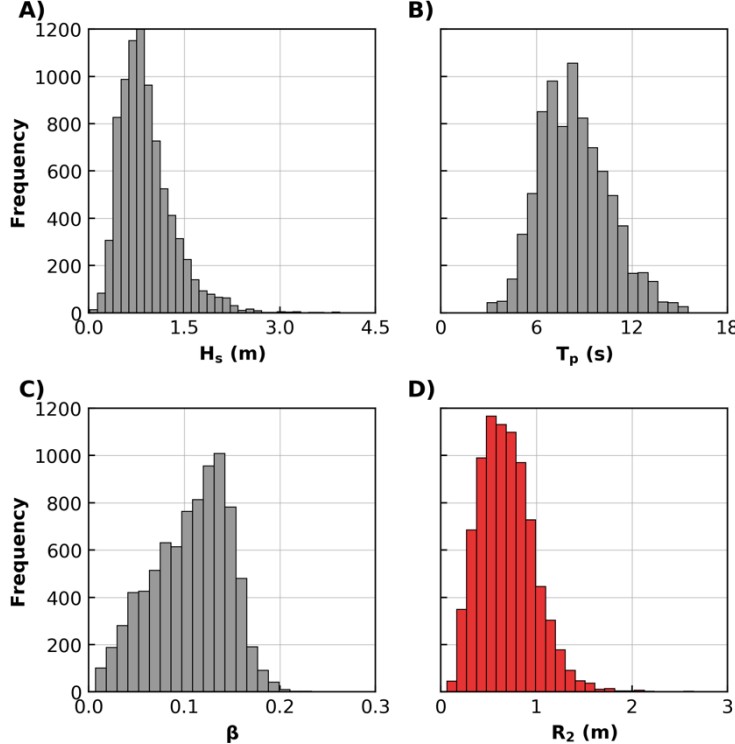


**Fig. 3: Histograms of the 8328 data samples extracted from the Narrabeen LIDAR: (A) significant wave height ($H_s$); (B) peak wave period ($T_p$); (C) beach slope ($\beta$); and, (D) 2% wave runup elevation ($R_2$).**

## 3.2    Training Data for the GP Runup Predictor

To determine the optimum training set size, kernel and model hyperparameters, a number of different user-defined training set sizes were trialed using the MDA selection routine discussed in **Sect. 2.3**. The GP was trained using different amounts of data and hyperparameters were optimized on the validation data set only. It was found that a training set size of only 5% of the available dataset (training dataset = 416 of 8328 available samples, validation dataset = 3956 samples, testing dataset = 3956 samples) was required to develop an optimum GP model. Training data sizes beyond this value produced negligible changes in GP performance but considerable increases in computational demand, similar to findings of previous work (Goldstein and Coco, 2014; Tinoco et al., 2015). Results presented below discuss the performance of the GP on the testing dataset which was not used in GP training or validation.





## 3.3 Runup Predictor Results

Results of the GP $R_2$ predictor on the 3956 test samples are shown in **Fig. 4**. This figure plots the mean GP predictions against corresponding observations of $R_2$. The mean GP prediction performs well on the test data, with a root-mean-squared-error (RMSE) of 0.18 m and bias (B) of 0.02 m. For comparison, the commonly used $R_2$ parameterization of Stockdon et al. (2006) tested on the same data has a RMSE of 0.36 m and B of 0.21 m. Despite the relatively accurate performance of the GP on this dataset, there remains significant scatter in the observed versus predicted $R_2$ in **Fig. 4**. This is consistent with recent work by Atkinson et al. (2017) showing that commonly used predictors of $R_2$ always result in scatter.

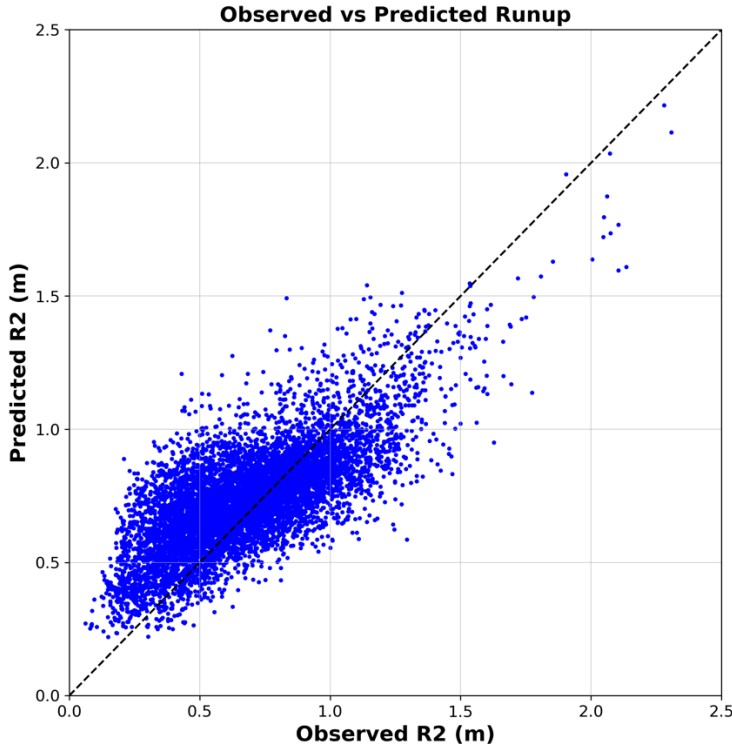

**Fig. 4: Observed 2% wave runup ($R_2$) versus the $R_2$ predicted by the Gaussian process model. Root-mean-squared-error (RMSE) is 0.36 m, bias (B) is 0.02 m and squared correlation ($r^2$) is 0.54.**

Here the scatter (uncertainty) is used to form ensemble predictions. The GP developed here not only gives a mean prediction as used in **Fig. 4**, but it specifies a multivariate Gaussian distribution from which different random functions that describe the data can be sampled. Random samples of wave





runup from the GP can capture uncertainty around the mean runup prediction (as was demonstrated in
the hypothetical example in **Fig. 1D**). To assess how well the GP model captures uncertainty, random
samples are successively drawn from the GP and the number of $R_2$ measurements captured with each
new draw are determined. Only 10 random samples drawn from the GP are required to capture 95% of
the scatter in $R_2$ (**Fig. 5A**). This process of drawing random samples from the GP was repeated 100
times with results showing that the above is true for any 10 random samples, with an average capture
percentage of 95.7% and range of 94.9% to 96.1% for 10 samples across the 100 trials. As a point of
contrast, **Fig. 5B** shows how much arbitrary error would need to be added to the mean $R_2$ prediction to
capture scatter about the mean to emulate the uncertainty captured by the GP. It can be seen that the
mean $R_2$ prediction would need to vary by ± 51% to capture 95% of the scatter present in the runup
data. This demonstrates how random models of runup drawn from the GP effectively capture
uncertainty in $R_2$ predictions. These randomly drawn $R_2$ models can be used within a larger dune-impact
model to produce an ensemble of dune erosion predictions that includes uncertainty in runup
predictions, as demonstrated in **Sect. 4**.


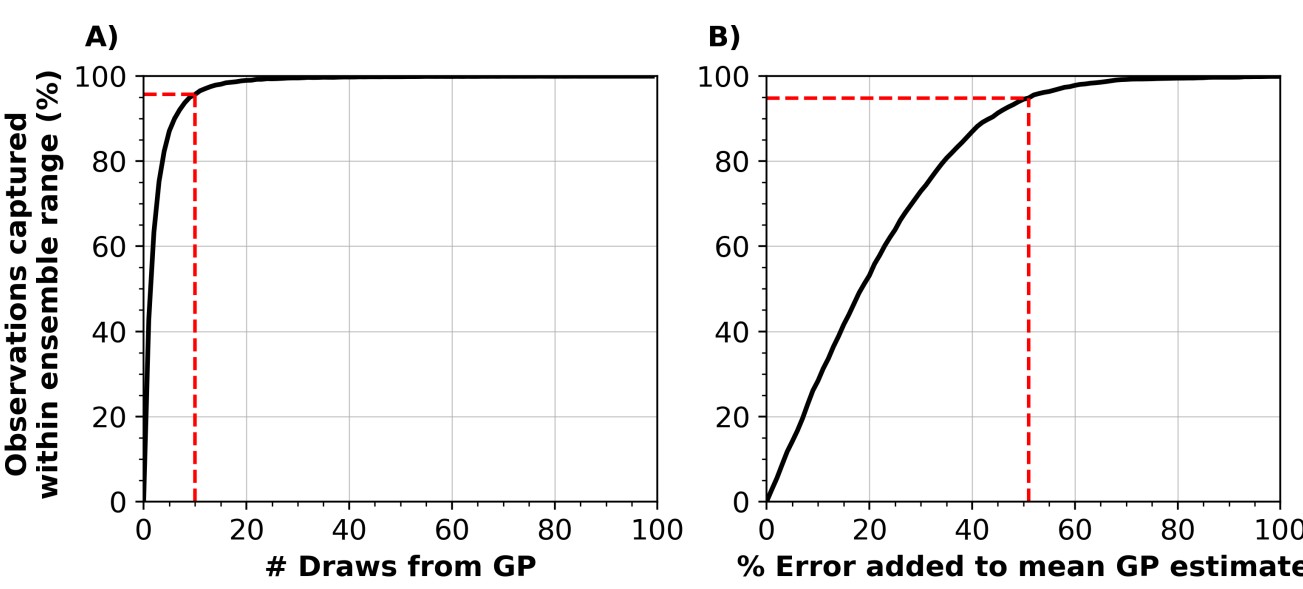




Fig. 5: A) Percent of observed runup values captured within the range of ensemble predictions made by randomly sampling different runup values from the Gaussian process. Only 10 randomly drawn models can form an ensemble that captures 95% of the scatter in observed $R_2$ values. B) An experiment showing how much arbitrary error would need to be added to the mean GP runup prediction in order to capture scatter in $R_2$ observations. The mean GP prediction would have to vary by 51% in order to capture 95% of scatter in $R_2$ observations.



## 4   Application of a Gaussian Process Runup Predictor in a Coastal Dune Erosion Model

### 4.1   Dune Erosion Model

We use the dune erosion model of Larson et al. (2004) as an example of how the GP runup predictor can be used to create an ensemble of dune erosion predictions, and thus provide probabilistic outcomes with uncertainty bands needed in coastal management. The dune erosion model is subsequently referred to as LEH04 and is defined as follows:

$$dV = 4C_s(R_2 - z_b)^2\left(\frac{t}{T}\right) \tag{9}$$

Where $dV$ (m$^3$/m) is the volumetric dune erosion per unit width alongshore for a given time step $t$, $z_b$ (m) is the time-varying dune toe elevation, $T$ (s) is the wave period for that time step, $R_2$ (m) is the 2% runup exceedance for that time step, and $C_s$ is the transport coefficient. Note that the original equation used a best-fit relationship to define the runup term, $R$ (see Eq. (36) in Larson et al., 2004) rather than $R_2$. Subsequent modifications of the LEH04 model have been made to adjust the collision frequency (i.e. the t/T term; e.g., Palmsten and Holman (2012), Splinter and Palmsten (2012)), however we retain the model presented in **Eq. (9)** for the purpose of providing a simple illustrative example. At each time step, dune volume is eroded in bulk and the dune toe is adjusted along a predefined slope (defined here as the linear slope between the pre- and post-storm dune toe) so that erosion causes the dune toe to increase in elevation and recede landward. Dune erosion and dune toe recession only occurs when wave runup ($R_2$) exceeds the dune toe (i.e., $R_2 - z_b > 0$) and cannot progress vertically beyond the maximum runup elevation. When $R_2$ does not exceed z$_b$, $dV = 0$. The GP $R_2$ predictor described in **Sect. 3** is used to stochastically parameterize wave runup in the LEH04 model and form ensembles of dune erosion predictions. The model is applied to new data not used to train the GP $R_2$ predictor, using detailed observations of dune erosion caused by a large coastal storm event at Narrabeen Beach, southeast Australia in 2011.





## 4.2    June 2011 Storm Data

In June 2011 a large coastal storm event impacted the southeast coast of Australia. This event resulted in variable alongshore dune erosion at Narrabeen Beach, which was precisely captured by airborne LIDAR immediately pre-, during, and post-storm by five surveys conducted approximately 24 hours apart. Cross-shore profiles were extracted from the Lidar data at 10 m alongshore intervals as described in detail in Splinter et al. (2018), resulting in 351 individual profiles (**Fig. 6**). The June 2011 storm lasted 120 hours. Hourly wave data was recorded by the Sydney wave rider buoy located in ~80 m water depth directly to the southeast of Narrabeen Beach. As with the hourly wave data used to develop the GP model of $R_2$ (**Sect. 3.1**), hourly wave data for each of the 351 profiles for the June 2011 storm was obtained by first transforming offshore wave data to the nearshore equivalent at 10 m water depth directly offshore of each profile using the SWAN spectral wave model (Booij et al., 1999), and then reverse shoaling back to equivalent deep water wave data, to account for the effects of wave refraction and attenuation caused by the distinctly curved Narrabeen embayment. The tidal range during the storm event was measured in-situ at the Fort Denison Tide Gauge (located within Sydney Harbour approximately 16 km south of Narrabeen) as 1.58 m (mean spring tidal range at Narrabeen is 1.6 m). The hydrodynamic time series and airborne LIDAR observations of dune change are used to demonstrate how the LEH04 model can be used with the GP predictor of runup to generate stochastic parameterizations and create probabilistic model ensembles (**Eq. (9)**).



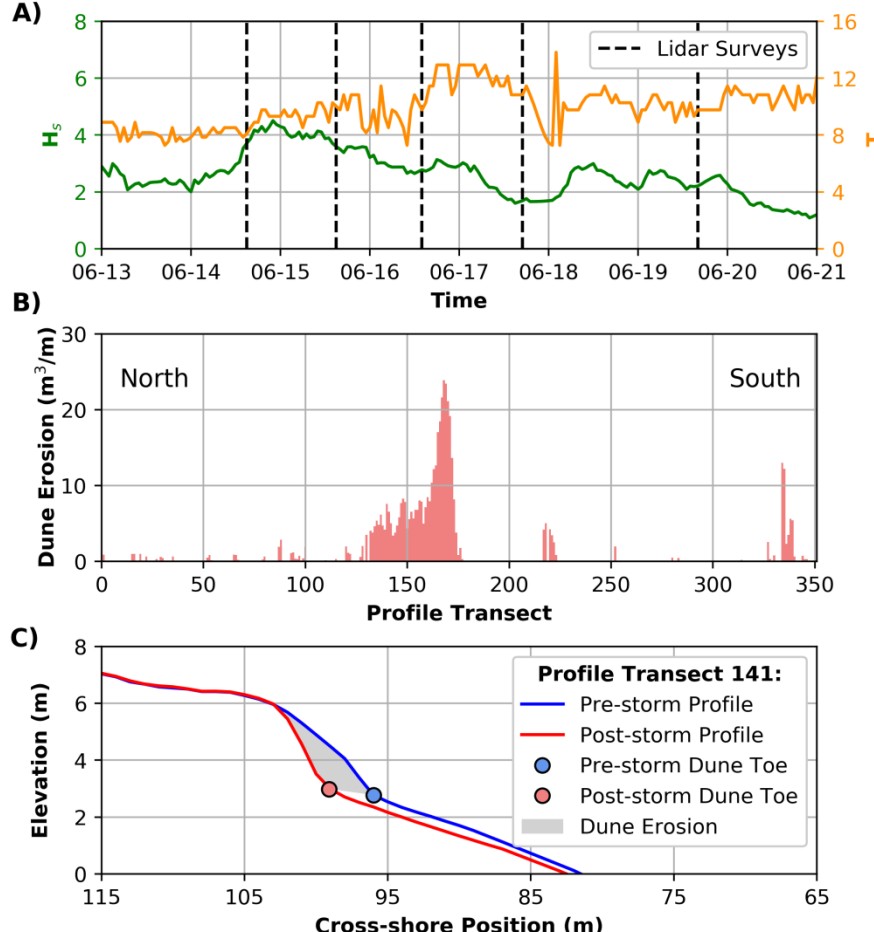

**Fig. 6: June 2011 storm data. A) Offshore $H_s$ and $T_p$ with vertical dashed lines indicating the time of the LIDAR surveys, B) Measured (pre vs post storm) dune erosion volumes for the 351 profile transects extracted from LIDAR data, C) Example pre- (blue) and post-storm (red) profile cross sections showing dune toes (coloured circles) and dune erosion volume (grey shading).**

For each of the 351 available profiles, the pre-, during and post-storm dune toe positions were defined as the local maxima of curvature of the beach profile following the method of Stockdon et al. (2007). Dune erosion at each profile was then defined as the difference in subaerial beach volume landward of the pre-storm dune toe, as shown in **Fig. 6C**. Of the 351 profiles, only 117 had storm driven dune erosion (**Fig. 6B**). For the example demonstration presented here, only profiles for which the post-storm dune toe elevation was at the same or higher elevation than the pre-storm dune toe are considered; which is a basic assumption of the LEH04 model. Of the 117 profiles with storm erosion, 40 profiles





met these criteria. For each of these profiles, the linear slope between the pre- and post-storm dune toe

was used to project the dune erosion calculated using the LEH04 model.

The LEH04 dune erosion model (**Eq. (9)**) has a single tuneable parameter, the transport coefficient $C_s$.

There is ambiguity in the literature regarding the value of $C_s$. Larson et al. (2004) developed an

empirical equation to relate $C_s$ to wave height ($H_{rms}$) and grain size ($D_{50}$) using experimental data.

Values ranged from $1 \times 10^{-5}$ to $1 \times 10^{-1}$, and Larson et al. (2004) used $1.7 \times 10^{-4}$ based on field data from

Birkemeier et al. (1988). Palmsten and Holman (2012) used LEH04 to model dune erosion observed in

a large wave tank experiment conducted at the O.H. Hinsdale Wave Research Laboratory at Oregon

State University. The model was shown to accurately reproduce dune erosion when applied in hourly

time steps using a $C_s$ of $1.34 \times 10^{-3}$, based on the empirical equation determined by Larson et al. (2004).

Mull and Ruggiero (2014) used values of $1.7 \times 10^{-4}$ and $1.34 \times 10^{-3}$ as lower and upper bounds of $C_s$ to

model dune erosion caused by a large storm event on the Pacific Northwest Coast of the USA and the

laboratory experiment used by Palmsten and Holman (2012). For the dune erosion experiment, the

value of $1.7 \times 10^{-4}$ was found to predict dune erosion volumes closest to the observed erosion when

applied in a single time step, with an optimum value of $2.98 \times 10^{-4}$. Splinter and Palmsten (2012) found

a best fit $C_s$ of $4 \times 10^{-5}$ in an application to modelling dune erosion caused by a large storm event that

occurred on the Gold Coast, Australia. Ranasinghe et al. (2012) found a $C_s$ value of $1.5 \times 10^{-3}$ in an

application at Narrabeen Beach, Australia. It is noted that $C_s$ values in these studies are influenced by

the time step used in the model and the exact definition of wave runup, $R$, used (Larson et al., 2004;

Mull and Ruggiero, 2014; Palmsten and Holman, 2012; Splinter and Palmsten, 2012). In practice, $C_s$

could be optimized to fit any particular dataset. However, for predictive applications the optimum $C_s$

value may not be known in advance, since it is unclear if subsequent storms at a given location will be

well predicted using previously optimized $C_s$ values. A key goal of this work is to determine if using

stochastic parameterizations to generate ensembles that predict a range of dune erosion (based on

uncertainty in the runup parameterization) can still capture observed dune erosion even if the optimum

$C_s$ value is not known in advance. As such, a $C_s$ value of $1.5 \times 10^{-3}$ is used in this example application





based on previous work at Narrabeen Beach by Ranasinghe et al. (2012). Sensitivity of model results to
the choice of $C_s$ are further discussed in **Sect. 5.2**.

An example at a single profile (profile 141, located approximately half-way up the Narrabeen
embayment as shown in **Fig. 6B**) of time-varying ensemble dune erosion predictions is provided in **Fig.**
**7**. It was previously shown in **Fig. 5** that only 10 random samples drawn from the GP $R_2$ predictor were
required to capture 95% of the scatter in the $R_2$ data used to develop and test the GP. However, it is
trivial to draw many more samples than this from the GP - for example, drawing 10,000 samples takes
less than one second on a standard desktop computer. Therefore, to explore a large range of possible
runup scenarios during the 120-hour storm event, 10,000 different runup time series are drawn from the
GP and used to run LEH04 at hourly intervals, thus producing 10,000 model results of dune erosion.
The effect of using different ensemble sizes is explored in **Sect. 5.2**. **Fig. 7A** shows the time-varying
distribution of the runup models (blue) used to force LEH04 along with the time-varying prediction
distribution of dune toe elevations (grey) throughout the 120-hour storm event. To interpret model
output probabilistically, the mean of the ensemble is plotted, along with intervals capturing 66%, 90%,
and 99% of the ensemble output. These intervals are consistent with those used in IPCC for climate
change predictions (Mastrandrea et al., 2010) and in the context of the model results presented here,
they represent varying levels of confidence in the model output. For example, there is high confidence
that the real dune erosion will fall within the 66% ensemble prediction range.  **Fig. 7B** shows the time-
varying predicted distribution of dune erosion volumes from the 10,000 LEH04 runs. It can be seen that
while the mean value of the ensemble predictions deviates slightly from the observed dune erosion, the
observed erosion is still captured well within the 66% envelope of predictions.





**Fig. 7: Example of LEH04 used with the Gaussian process $R_2$ predictor to form an ensemble of dune erosion predictions. 10,000 runup models are drawn from the Gaussian process and used to force the LEH04 model. A) Runup (blue) and dune toe (grey) elevation for the 120-hour storm event. Bold colored line is the mean of the ensemble and shaded areas represent the regions captured by 66%, 90% and 99% of the ensemble predictions. Pink dots denote the observed dune toe elevation throughout the storm event. B) The corresponding ensemble of dune erosion predictions.**

Pre- and post-storm dune erosion results for the 40 profiles using 10,000 ensemble members and $C_s$ of 1.5 x 10$^{-3}$ are shown in **Fig. 8**. The squared-correlation ($r^2$) for the observed and predicted dune erosion volumes is 0.85. Many of the profiles experienced only minor dune erosion (< 2.5 m$^3$/m) and can be seen to be well predicted by the mean of the ensemble predictions. In contrast, the ensemble mean can be seen to under-predict dune erosion at profiles where high erosion volumes were observed. However, the ensemble range of predictions for these profiles also has a large spread, indicative of high uncertainty in predictions. It should be noted that the results presented in **Fig. 8** are based on a non-optimized $C_s$ value. Increasing $C_s$ would lead to better mean ensemble predictions of the large dune

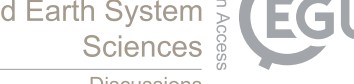



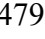

erosion volumes, but potentially over-prediction of the smaller events. The exact effect of varying $C_s$ is
quantified in **Sect. 5.2**. However, regardless of the value of $C_s$ chosen, an advantage of the GP approach
is that uncertainty in the GP predictions can give an indication of dune erosion, even if the mean dune
erosion prediction deviates from the observation.

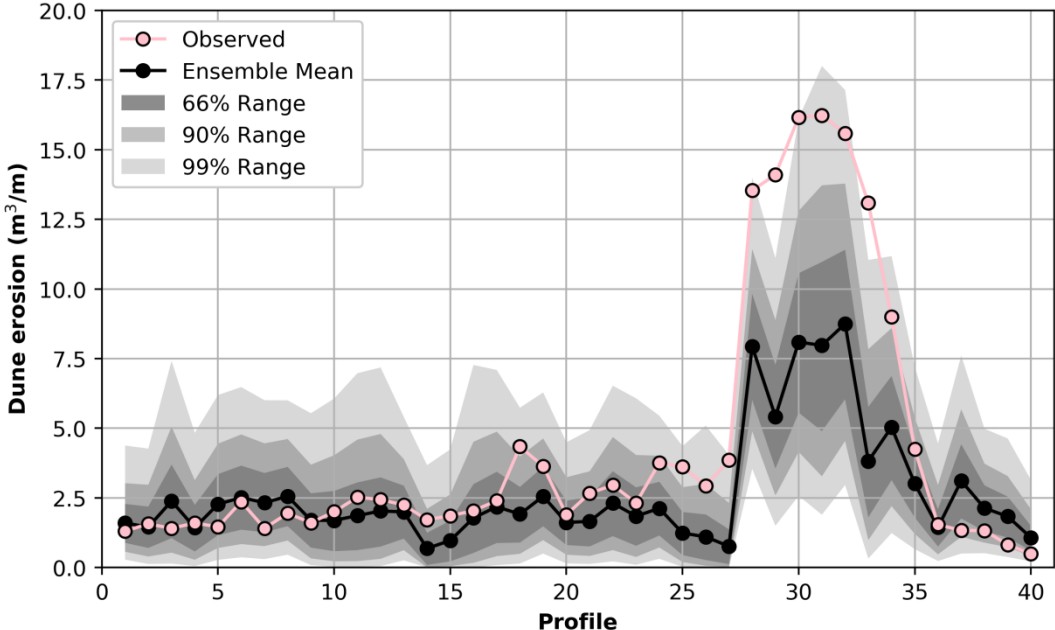


Fig. 8: Observed (pink dots) and predicted (black dots) dune erosion volumes for the 40 modelled profiles, using 10,000 runup
models drawn from the Gaussian process and used to force the LEH04 model. Note that the 40 profiles shown are not uniformly
spaced along the 3.5 km Narrabeen embayment. The black dots represent the ensemble mean prediction for each profile, while the
shaded areas represent the regions captured by 66%, 90% and 99% of the ensemble predictions.



## 5   Discussion

### 5.1   Runup Predictors

Studies of commonly used deterministic runup parameterizations such as those proposed by Hunt (1959), Holman (1986) and Stockdon et al. (2006) amongst others, show that these parametrizations are not universally applicable and there remains no perfect predictor of wave runup on beaches (Atkinson et al., 2017; Passarella et al., 2018a; Power et al., 2018). This suggests that the available parametrizations do not fully capture all the relevant processes controlling wave runup on beaches (Power et al., 2018). Recent work has used ensemble and data-driven methods to account for unresolved factors and complexity in runup processes. For example, Atkinson et al. (2017) developed a 'model-of-models' by fitting a least-squares line to the predictions of several runup parameterizations. Power et al. (2018) used a data-driven, deterministic, Gene-Expression Programming model to predict wave runup against a large dataset of runup observations. Both of these approaches led to improved predictions, when compared to conventional runup parameterizations, of wave runup on the datasets tested in these studies. The work presented in this study used a data-driven Gaussian process (GP) approach to develop a probabilistic runup predictor. While the mean predictions from the GP predictor developed in this study using high-resolution LIDAR data of wave runup were accurate (RMSE = 0.18 m) and better than those provided by the Stockdon et al. (2006) formula tested on the same data (RMSE = 0.36 m), the key advantage of the GP approach over deterministic approaches is that probabilistic predictions are output that are specifically derived from data and implicitly account for unresolved processes and uncertainty in runup predictions. Previous work has similarly used GPs for efficiently and accurately quantifying uncertainty in other environmental applications (e.g., Holman et al., 2014; Kupilik et al., 2018; Reggente et al., 2014). While alternative approaches are available for generating probabilistic predictions, such as Monte Carlo simulations (e.g., Callaghan et al., 2013), the GP approach explicitly derives uncertainty from data, requires no deterministic equations, and is computationally efficient (i.e., as discussed in **Sect. 5.2**, drawing 10,000 samples of 120-hour runup time series on a standard desktop computer took less than one second).



## 5.2   The Effect of Cs and Ensemble Size on Dune Erosion

In **Sect. 4**, the application of the GP runup predictor within the LEH04 model to produce an ensemble of dune erosion predictions was based on 10,000 ensemble members and a $C_s$ value of 1.5 x $10^{-3}$. The sensitivity of results to the number of members in the ensemble and the value of the tunable parameter $C_s$ in **Eq. (9)** is presented in **Fig. 9.** The mean absolute error (MAE) between the mean ensemble dune erosion predictions and the observed dune erosion, averaged across all 40 profiles, varies for $R_2$ ensembles of 5, 10, 20, 100, 1000, and 10,000 members and $C_s$ values ranging from $10^{-5}$ to $10^{-1}$ (**Fig. 9).** As can be seen in **Fig. 9A** and summarized in **Table 1**, the lowest MAE for the differing ensemble sizes is similar, ranging from 1.50 to 1.64 m³/m, suggesting that the number of ensemble members does not have a significant impact on the resultant mean prediction. The lowest MAE for the different ensemble sizes corresponds to $C_s$ values between 2.8 x $10^{-3}$ (10,000 ensemble members) and 4.1 x $10^{-3}$ (5 ensemble members); reasonably consistent with the value of 1.5 x $10^{-3}$ previously reported by Ranasinghe et al. (2012) for Narrabeen Beach and within the range of $C_s$ values presented in Larson et al. (2004).

The key utility to using a data-driven GP predictor to produce ensembles is that a range of predictions at every location is provided as opposed to a single erosion volume. The ensemble range provides an indication of uncertainty in predictions, which can be highly useful for coastal engineers and managers taking a risk-based approach to coastal hazard management. **Fig. 9B-D** displays the percentage of dune erosion observations from the 40 profiles captured within ensemble predictions for $C_s$ values ranging from $10^{-5}$ to $10^{-1}$. It can be seen that a high proportion of dune erosion observations are captured within the 66%, 90% and 99% ensemble envelope across several orders of magnitude $C_s$. While the main purpose of using ensemble runup predictions within LEH04 is to incorporate uncertainty in the runup prediction, this result demonstrates that the ensemble approach is less sensitive to the choice of $C_s$ than a deterministic model and so can be useful for forecasting with non-optimized model parameters.

Results in **Fig. 9** and **Table 1** demonstrate that there is relatively little difference in model performance when more than 10 to 100 ensemble members are used; consistent with results presented previously in



**Fig. 5** that showed that only 10 random samples drawn from the GP $R_2$ predictor were required to
capture 95% of the scatter in the $R_2$ data used to develop and test the GP. This suggests that the GP
approach efficiently captures scatter (uncertainty) in runup predictions and subsequently, dune erosion
predictions, requiring on the order of 10 samples; significantly less than the $10^3 - 10^6$ runs typically
used in Monte Carlo simulations to develop probabilistic predictions (e.g., Callaghan et al., 2008; Li et
al., 2013; Ranasinghe et al., 2012). Nevertheless, it is noted that drawing a large number of samples
from the GP predictor is trivial, with 10,000 samples taking less than one second on a standard desktop
computer.





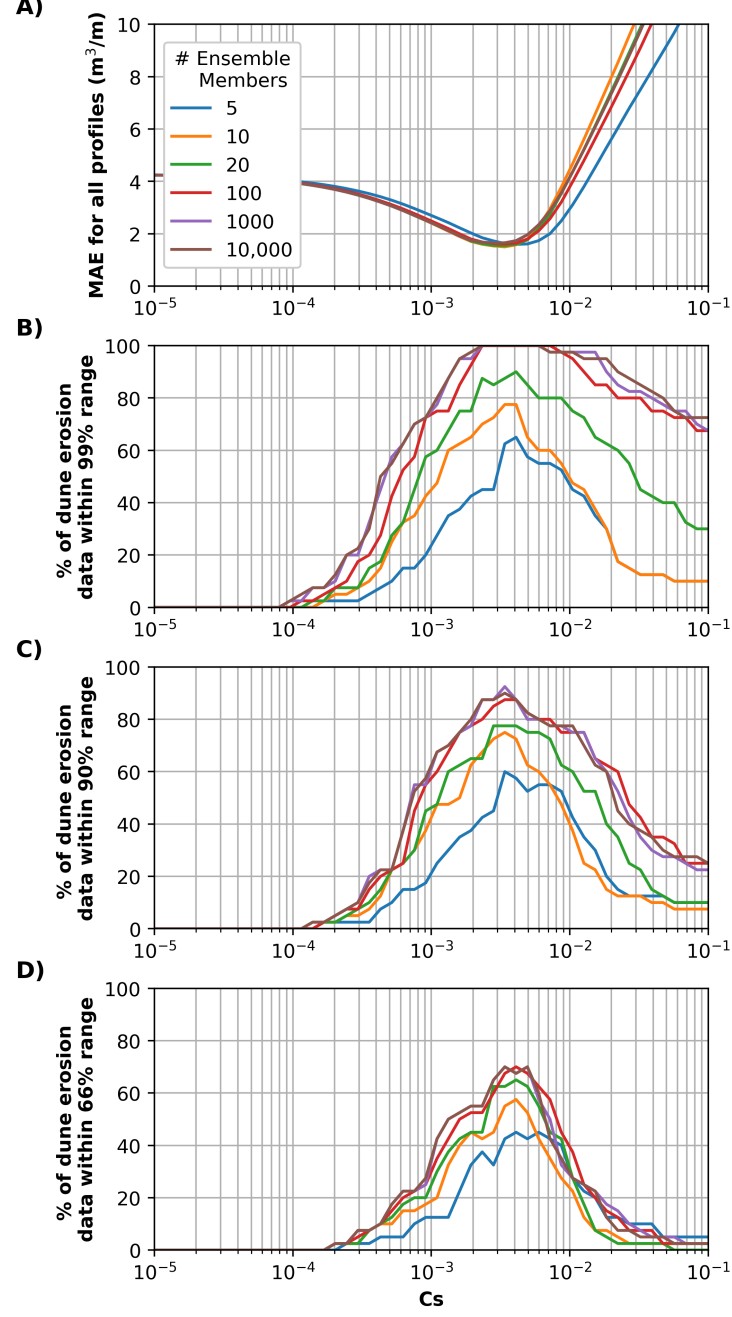


**Fig. 9: Results of the stochastic parameterization methodology for $R_2$ ensembles of 5, 10, 20, 100, 1000, and 10,000 members and $C_s$ values ranging from $10^{-5}$ to $10^{-1}$. A) The mean absolute error (MAE) between the median ensemble dune erosion predictions and the observed dune erosion averaged across all 40 profiles. B), C) and D) show the percentage of dune erosion observations that fall within the 99%, 90% and 66% ensemble prediction ranges respectively.**



**Table 1: Quantitative summary of Fig. 9, showing the optimum $C_s$ value for differing ensemble sizes, along with the associated**
**mean-absolute-error (MAE) and percent of the 40 dune erosion observations captured by the 66%, 90% and 99% ensemble**
**prediction range.**

| Ensemble Members | Optimum $C_s$ | MAE (m³/m) | r² | Percent Captured in 66% Ensemble Range (%) | Percent Captured in 90% Ensemble Range (%) | Percent Captured in 99% Ensemble Range (%) |
|---|---|---|---|---|---|---|
| 5 | $4.1 \times 10^{-3}$ | 1.59 | 0.86 | 45 | 57 | 65 |
| 10 | $3.4 \times 10^{-3}$ | 1.50 | 0.87 | 55 | 75 | 78 |
| 20 | $3.4 \times 10^{-3}$ | 1.54 | 0.86 | 62 | 78 | 88 |
| 100 | $3.3 \times 10^{-3}$ | 1.61 | 0.86 | 68 | 88 | 100 |
| 1000 | $2.8 \times 10^{-3}$ | 1.64 | 0.86 | 65 | 88 | 100 |
| 10,000 | $2.8 \times 10^{-3}$ | 1.64 | 0.86 | 65 | 88 | 100 |


**5.3    Including Uncertainty in Dune Erosion Models**
Uncertainty in wave runup predictions within dune-impact models can result in significantly varied
predictions of dune erosion. For example, the model of Larson et al. (2004) used in this study only
predicts dune erosion if runup elevation exceeds the dune toe elevation and predicts a non-linear
relationship between runup that exceeds the dune toe and resultant dune erosion. Hence, if wave runup
predictions are biased too low then no dune erosion will be predicted, and if wave runup is predicted too
high then dune erosion may be significantly over predicted. Ensemble modelling has become standard
practice in many areas of weather and climate modelling (Bauer et al., 2015), hydrological modelling
(Cloke and Pappenberger, 2009), and more recently has been applied to coastal problems such as the
prediction of cliff retreat (Limber et al., 2018) as a method of handling prediction uncertainty. While
using a single deterministic model is computationally simple and provides one solution for a given set
of input conditions, model ensembles provide a range of predictions that can better capture the variety
of mechanisms and stochasticity within a coastal system. The result is typically improved skill  over
deterministic models (Atkinson et al., 2017; Limber et al., 2018) and a natural method of providing
uncertainty with predictions.





As a quantitative comparison, Splinter et al. (2018) applied a modified version of the LEH04 model to
the same June 2011 storm dataset used in the work presented here with a modified expression for the
collision frequency (i.e. the t/T term in **Eq. (9)**) based on work by Palmsten and Holman (2012). The
parameterization of Stockdon et al. (2006) was used to estimate $R_2$ in the model. The model was forced
hourly over the course of the storm, updating the dune toe, recession slope, and profiles based on each
daily LIDAR survey. Based on only the 40 profiles used in the present study, results from Splinter et al.
(2018) showed that the deterministic LEH04 approach reproduced 68% ($r^2 = 0.68$) of the observed
variability in dune erosion. As shown in **Table 1**, the simple LEH04 model (**Eq. (9)**) applied here using
the GP runup predictor to generate ensemble prediction reproduced ~85% (based on the ensemble
mean) of the observed variability in dune erosion for the 40 profiles. While there are some discrepancies
in the two modelling approaches, the ensemble approach clearly has an appreciable increase in skill
over the deterministic approach; attributed here to using a runup predictor trained on local runup data,
and the ensemble modelling approach. However, a major advantage of the ensemble approach over the
deterministic approach is the provision of prediction uncertainty (e.g., **Fig. 8**). While the mean ensemble
prediction is not 100% accurate, **Table 1** shows that using just 100 samples can capture all the observed
variability in dune erosion within the ensemble output.

The GP approach is a novel approach to building model ensembles to capture uncertainty. Previous
work modelling beach and dune erosion has successfully used Monte Carlo methods, which randomly
vary model inputs within many thousands of model iterations, to produce ensembles and probabilistic
erosion predictions (e.g., Callaghan et al., 2008; Li et al., 2013; Ranasinghe et al., 2012). As discussed
earlier in **Sect. 5.2**, advantages of the GP approach over approaches like Monte Carlo include the
explicit quantification of uncertainty directly from data, no deterministic equations are required, and the
approach is computationally efficient; here, drawing 10,000 samples of 120-hour runup time series from
the GP took less than one second on a standard desktop computer.



## 6 Conclusion


For coastal managers, the accurate prediction of wave runup as well as dune erosion is critical for
characterizing the vulnerability of coastlines to wave-induced flooding, erosion of dune systems, and
wave impacts on adjacent coastal infrastructure. While many formulations for wave runup have been
proposed over the years, none have proven to accurately predict runup over a wide range of conditions
and sites of interest. In this contribution, a Gaussian process (GP) was used with over 8000 high-
resolution LIDAR-derived wave runup observations were used to develop a probabilistically
parametrization of wave runup that quantify uncertainty in runup predictions. The mean GP prediction
performed well on unseen data, with a RMSE of 0.18 m, a significant improvement over the commonly
used $R_2$ parameterization of Stockdon et al. (2006) (RMSE of 0.36 m) used on the same data. Further,
only 10 randomly drawn models from the probabilistic GP distribution were needed to form an
ensemble that captured 95% of the scatter in the test data.

Coastal dune-impact models offer a method of predicting dune erosion deterministically. As an example
application of how the GP runup predictor can be used in geomorphic systems, the uncertainty in the
runup parameterization was propagated through a deterministic dune erosion model to generate
ensemble model predictions and provide prediction uncertainty. The hybrid dune erosion model
performed well on the test data, with a squared-correlation ($r^2$) between the observed and predicted dune
erosion volumes of 0.85. Importantly, the probabilistic output provided uncertainty bands of the
expected erosion volumes which is a key advantage over deterministic approaches. Compared to
traditional methods of producing probabilistic predictions such as Monte Carlo, the GP approach has the
advantage of learning uncertainty directly from observed data, it requires no deterministic equations,
and is computationally efficient; for the GP developed here, drawing 10,000 samples of 120-hour runup
time series on a standard desktop computer took less than one second.

This work is an example of how a machine learning model such as a GP can profitably be integrated
into coastal morphodynamic models (Goldstein and Coco, 2015) to provide probabilistic predictions for
nonlinear, multidimensional processes and drive ensemble forecasts. Approaches combining machine



learning methods with traditional coastal science and management models present a promising area for
furthering coastal morphodynamic research. Future work is focused on using more and varied datasets
to further train the GP developed here and to integrate it into a real-time coastal erosion forecasting
system.





## Code and Data Availability

The data and code used to develop the Gaussian Process runup predictor in this manuscript are publicly available at https://github.com/TomasBeuzen/BeuzenEtAl_GP_Paper.





## 632 **Author Contributions**

633 The order of the authors' names reflects the size of their contribution to the writing of this manuscript.



## Acknowledgements

This research was partially funded by ongoing support by Northern Beaches Council, the Australian Research Council (LP04555157, LP100200348, DP150101339) and the NSW Environmental Trust Environmental Research Program (RD 2015/0128). Wave and tide data were kindly provided by Manly Hydraulics Laboratory under the NSW Coastal Data Network Program managed by OEH. The lead Author is funded under the Australian Postgraduate Research Training Program. EBG acknowledges financial support from DOD DARPA (R0011836623/HR001118200064).



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
