# Peer review of "Ensemble models from machine learning: an example of wave runup"

_Natural Hazards and Earth System Sciences, 2019_

## Referee Comment (RC1) · Anonymous Referee #1 · 30 Apr 2019

General Comments:

Beuzen et al., present a probabilistic, data-driven model for ensemble predictions of wave runup and dune erosion. They show their model has a lower RMSE and bias than other frequently used, deterministic runup predictions, and has the ability to produce confidence bounds on runup (and thus dune erosion) predictions. This technique could bring value to erosion forecasting as well as to the development of probabilistic hazard zones. As such, I feel that the research forms an important contribution. The manuscript is well written and describes the motivation, model, and results very clearly. However, I feel the authors need to discuss the limitations of their approach more outright as well as improve some of the structure of the results and discussion sections. Nevertheless, if these comments are adequately addressed, I find that this

paper would be a valuable addition to the literature.

Specific Comments:

As mentioned above, this manuscript develops a novel method for including uncertainties within estimates of wave runup, an important contributor to coastal erosion hazards. While there is certainly merit to the results presented, I feel that the manuscript would benefit from a discussion of the limitations and assumptions of the modeling technique as well as some changes to the overall structure.

My main concern is that the authors neglect to discuss the limitations of their methodology. For example, the authors' state that machine learning models perform poorly when forced to extrapolate a prediction, and that it is important to use training data sets that capture the full range of variability of the data. While the authors use training data from a year, this may not account for interannual variability of the wave climate and long-term trends or shifts in storm tracks and intensities. The authors should discuss how representative the year of training data is of past years, wave climate-wise, or other years of measurements, runup-wise, as well as the precautions in a limited set of training data. Furthermore, rare, extreme events may not have occurred during that particular year. This is often also one of the issues with deterministic/empirical parameterizations of the R2% from field studies – that extreme conditions haven't been captured. On that note, the authors seem to suggest in their discussion that this technique is more reliable than Monte Carlo techniques because the uncertainty is learned directly from data. Monte Carlo techniques however, seek to represent conditions that haven't necessarily occurred by modeling large sets of physically plausible conditions. Both the GP and MC techniques seem equally useful, with different motivations.

Following on, there is no adequate discussion of the model's poor prediction for large erosion events seen in Figure 8 between profiles 28 -34. The authors suggest that the under-prediction of large erosion events could be due to a non-optimized Cs parameter, stating that, "…increasing Cs would lead to better mean ensemble predictions of

the large dune erosion volumes but over-prediction of the smaller events." Does this mean that the optimum Cs within table 1 for each ensemble grouping captured erosion over this set of profiles better? Is it more important to weight the mean or extreme conditions?

Then, in Lines 534-535, "the results demonstrate the ensemble approach is less sensitive to the choice of Cs than a deterministic model so it can be useful for forecasting with non-optimized model parameters". So on one hand, they suggest that erosion can be better predicted with an optimized Cs, and on the other hand it will be useful for forecasting with non-optimized model parameters. Can the authors please expand on these seemingly contradictory statements?

In Figure 8, there are some points that fall well outside of the range of uncertainty in erosion measurements. The authors state that, "regardless of the value of Cs chosen, an advantage of the GP approach is that uncertainty in GP predictions can give an indication of dune erosion, even if the mean dune erosion prediction deviates from the observations. " While there is truth to this statement, there are locations where the uncertainty does not characterize the observations at all, and this should be discussed.

Furthermore, I understand that the main contribution of the paper is the ensemble technique for modeling wave runup rather than the erosion model calibration, however I found the sections focused on the transport coefficient, Cs, to break up the flow of the manuscript. Specifically, the section describing previous research on Cs (Lines 415 – 440) can be shortened and put into an appendix or supplemental information so the reader can focus instead on the results. I felt the most important information in this section was what Cs value was being used, how much it ranged, and specifically Lines 430 – 440. Additionally, the results that appear in the discussion section (section 5.2) could be moved to the end of the results section. While these results are important for weighing the importance of Cs, they felt out of place in the discussion section.

Finally, this manuscript left me wondering what type of applicability this has to other

locations. For example, is this method limited to locations with data?

Comments on Specific Lines:

Lines 294 – 295: The authors may want to consider citing the technical methods for extracting wave runup as this is not a trivial task.

Lines 392 – 393: What is the resolution of SWAN model? 10m?

Line 316: does "this value" refer to 5% or N = 416?

Figures:

Figure 3: Why is the wave runup histogram plotted in red?

Figure 7a: Does the blue display the runup prediction or the total water level prediction? It looks to me like there are tides within the blue values. Also, is it possible to plot some of the wave runup data on 7a? Authors show observations of erosion and dune toe, however, their model is predicting wave runup so that would be interesting to see in the time series sense too.

Technical Corrections:

Line 35: The authors use the term "significant wave period" to describe the wave period variable in runup formulations and then later mostly use "peak wave period." As "significant wave period" is not used as typically, I'd recommend sticking with "peak wave period" or simply "wave period."

Line 285: Please define SSE acronym, and if not used again, no acronym is necessary.

Lines 446 – 447, Lines 509-510, Lines 545-546, Lines 595 – 596, Lines 619-620: repetition of similar variations of the following text, "drawing 10,000 samples takes than one second on a standard desktop computer." I'd recommend the authors say this a few times, then perhaps the term "computationally efficient" as this felt repetitive to read 5 times.

Line 603: remove "were used"

---

## Referee Comment (RC2) · Anonymous Referee #2 · 27 May 2019

Review of: Ensemble models from machine learning: an example of wave runup and coastal dune erosion

By Thomas Beuzen, Evan B. Goldstein, and Kristen D. Splinter

I have reviewed the above manuscript and find that it will be acceptable for publication in Natural Hazards and Earth System Sciences following only a very few very minor revisions. The paper "Ensemble models from machine learning: an example of wave runup and coastal dune erosion" uses a machine learning technique, Gaussian Process Regression, to develop a probabilistic wave runup model able to be implemented in an ensemble approach. The wave runup model is then applied to a deterministic dune erosion model to demonstrate the power of hybrid approaches over typical deterministic approaches. This topic is of considerable importance, is definitely appropriate

for this journal, and will be useful to a broad audience. I have only a couple of general and specific comments that might aid in improving the manuscript and these are offered below. This manuscript is extremely well written and therefore I am not submitting an annotated version of the manuscript.

General Comments: The authors make the bold (and most likely correct) statement that the development of a perfect deterministic parameterization of wave runup using only the typical inputs of beach slope, wave height, and wave period is improbable. They then go on to develop a GP runup model that has higher skill than the most typical deterministic runup model used today (Stockdon et al., 2006). However, to build this new model they still use the same three easily obtainable inputs. While perfectly reasonable for this paper's demonstration purposes, I am left wondering whether or not GP could be used to build an even better runup model if other input forcing dimensions were included? Figure 4 appears to have some structure in it, with low values of R2 overpredicted and high values underpredicted. Can we learn something from this? Even a few suggestions and/or speculations from the authors would be welcome about machine learning directions for developing even better runup models. In developing the input Hs and Tp time series for both the creation of the runup model and for the ultimate test against the dune erosion event, it is mentioned that SWAN is used to transform all conditions into the nearshore before being linear back shoaled. Did the authors really run 100s to 1000s of individual SWAN simulations? This effort seems like it must have had a high computational cost? Since the paper emphasizes the efficiency of the GP runup model some more detail of this step in the process is warranted. Have the authors considered developing simple look up tables, or better yet, a GP model of SWAN to simplify this stage of the process? The decision to use MDA for developing the training data seems sound. However, a list, or discussion of other possible space filling algorithms might be useful for readers embarking on their own GP applications. I commend the authors for their relatively parsimonious and clear explanation of GP theory in section 2.1. However, I suspect that this treatment will still be an occasionally opaque to some readers (including this reviewer). My only

suggestion here is to continue to work on describing machine learning approaches such as GP in as clear of terms as possible. This paper does this as well as I have seen.

Specific Comments: Line 81-82: Maybe add to this growing body of literature by including: Parker, K., P. Ruggiero, K. Serafin, and D. Hill. 2019. "Emulation as an Approach for Rapid Estuarine Modeling." Coastal Engineering 150: 79–93. https://doi.org/10.1016/j.coastaleng.2019.03.004. Line 235-236: I thank the author for identifying which toolkit they used in developing the runup GP model. However, it might be helpful for a broad group of readers if the authors listed other potential toolkits that could also have been used – say for example in Matlab, or R?

Line 544-566: The statement about 10,000 samples taking less than one second on a standard desktop computer is repetitive at this point.

Thanks very much for the opportunity to review this very exciting manuscript.

---

## Author Comment (AC1) · 5 Aug 2019

Please see the attached .pdf file for a formatted version of the author response document.

We thank both Reviewers for their time and effort in providing constructive feedback on our manuscript. Their comments have led to a much-improved manuscript with greater clarity in the description and purpose of the work presented. Below is our point-by-point response to the comments made and details of where the related changes have been made in the revised manuscript, in which they have also been highlighted. For clarity, Reviewer comments have been separated into key points which are addressed individually.

[Figure]

REVIEWER 1

Comment 1: "My main concern is that the authors neglect to discuss the limitations of their methodology. For example, the authors' state that machine learning models perform poorly when forced to extrapolate a prediction, and that it is important to use training data sets that capture the full range of variability of the data. While the authors use training data from a year, this may not account for interannual variability of the wave climate and long-term trends or shifts in storm tracks and intensities. The authors should discuss how representative the year of training data is of past years, wave climate-wise, or other years of measurements, runup-wise, as well as the precautions in a limited set of training data. Furthermore, rare, extreme events may not have occurred during that particular year. This is often also one of the issues with deterministic/empirical parameterizations of the R2% from field studies – that extreme conditions haven't been captured."

Author Response to Comment 1: The June 2011 storm event used in the testing phase of this manuscript (Sect. 4.2) lies within the range of the one-year training dataset used to develop the GP runup predictor, so extrapolation is not an issue in the work presented. New Lines 401 – 403 have been added to the manuscript to clarify this: "As can be seen in Fig. 6 the wave conditions for the June 2011 storm lie within the range of the training dataset used to develop the GP runup predictor." However, the Reviewer raises the important issue that capturing the full range of variability in a dataset used to train a GP, or any machine learning model can be difficult, especially when considering longer-term trends and a potentially changing wave climate in the future. This limitation has now been clarified in the manuscript at Lines 564 – 569: "However, as discussed in Sect. 2.3, when developing a GP, or any machine learning model, the training data should include the full range of possible variability in the data to be modelled in order to avoid extrapolation. A limitation of using this data-driven approach for runup prediction is that it can be difficult to acquire a training dataset that capture all possible variability in the system, from, for example, longer-term trends, extreme events or a potentially

changing wave climate in the future (Semedo et al., 2012)."

Comment 2: "On that note, the authors seem to suggest in their discussion that this technique is more reliable than Monte Carlo techniques because the uncertainty is learned directly from data. Monte Carlo techniques however, seek to represent conditions that haven't necessarily occurred by modeling large sets of physically plausible conditions. Both the GP and MC techniques seem equally useful, with different motivations."

Author Response to Comment 2: We agree with the Reviewer that both GP and MC techniques have differing applications and advantages/limitations. It was not the intention of the manuscript to argue that GP is better than MC but simply to illustrate the GP approach and how it can be applied to coastal process prediction. We have made wording changes to clarify this at: Lines 561 – 562: "While alternative approaches are available for generating probabilistic predictions, such as Monte Carlo simulations (e.g., Callaghan et al., 2013), the GP approach offers a method of deriving uncertainty explicitly from data. . ." Line 607 – 608: "As discussed earlier in Sect. 4.3, the GP approach differs to Monte Carlo in that it explicitly quantifies uncertainty directly from data, does not use deterministic equations, and can be computationally efficient."

Comment 3: "Following on, there is no adequate discussion of the model's poor prediction for large erosion events seen in Figure 8 between profiles 28 -34. The authors suggest that the under-prediction of large erosion events could be due to a non-optimized $C_s$ parameter, stating that, ". . .increasing $C_s$ would lead to better mean ensemble predictions of C2 the large dune erosion volumes but over-prediction of the smaller events." Does this mean that the optimum $C_s$ within table 1 for each ensemble grouping captured erosion over this set of profiles better? Is it more important to weight the mean or extreme conditions? Then, in Lines 534-535, "the results demonstrate the ensemble approach is less sensitive to the choice of $C_s$ than a deterministic model so it can be useful for forecasting with non-optimized model parameters". So on one hand, they suggest that erosion can be better predicted with an optimized $C_s$, and on

the other hand it will be useful for forecasting with non-optimized model parameters. Can the authors please expand on these seemingly contradictory statements?"

Author Response to Comment 3: The Reviewer is correct that the Cs value used in Fig. 8 is non-optimized (it is an assumed value of 1.5 x 10-3, the point being that we often would not know the optimum Cs value in advance, particularly in a forecasting scenario), and that the Cs values in Table 1 are the actual optimized values which would fit this dataset better. Of course, the best prediction will always be achieved with an optimized Cs. However, because this optimum value cannot be known in advance, we are demonstrating here how a GP can provides useful insights about uncertainty even when based on a non-optimized Cs. This key point has been clarified with changes to Lines 480 – 487: "It should be noted that the results presented in Fig. 8 are based on an assumed (i.e., non-optimized) Cs value of 1.5 x 10-3. Better prediction of large erosion events could potentially be achieved by increasing Cs or giving greater weighting to these events during calibration, but at the cost of over-predicting the smaller events. The exact effect of varying Cs is quantified in Sect. 4.3. Importantly, Fig. 8 demonstrates that even with a non-optimized Cs, uncertainty in the GP predictions can provide useful information about the potential for dune erosion, even if the mean dune erosion prediction deviates from the observation; a key advantage of the GP approach over a deterministic approach."

Comment 4: "In Figure 8, there are some points that fall well outside of the range of uncertainty in erosion measurements. The authors state that, "regardless of the value of Cs chosen, an advantage of the GP approach is that uncertainty in GP predictions can give an indication of dune erosion, even if the mean dune erosion prediction deviates from the observations. " While there is truth to this statement, there are locations where the uncertainty does not characterize the observations at all, and this should be discussed."

Author Response to Comment 4: It is true that some of the dune erosion observations in Fig. 8 fall outside the range of uncertainty predicted by the GP (Profiles 29, 30 and

33). This is most likely due to the non-optimized Cs value used and/or inadequacies in the GP runup model or L04 dune erosion model. However, as stated in Lines 494 – 495, the range of ensemble predictions output by the GP at these particular profiles is also very large (much larger than then the range of ensemble predictions at Profiles 1 – 26), which is indicative of high uncertainty in predictions and the potential for high erosion to occur at these profiles. While the model is clearly not perfect, the idea is that using the GP runup predictor provides a useful indication of uncertainty in predictions – which is an advantage over the point prediction a deterministic approach would provide. Words to this effect have now been included in Lines 476–487: "In contrast, the ensemble mean can be seen to under-predict dune erosion at profiles where high erosion volumes were observed (profiles 29 – 34 in Fig. 8) with some profiles not even captured by the uncertainty of the ensemble. However, the ensemble range of predictions for these particular profiles also has a large spread, indicative of high uncertainty in predictions and the potential for high erosion to occur. . . Importantly, Fig. 8 demonstrates that even with a non-optimized Cs uncertainty in the GP predictions can provide useful information about the potential for dune erosion, even if the mean dune erosion prediction deviates from the observation; a key advantage of the GP approach over a deterministic approach."

Comment 5: "Furthermore, I understand that the main contribution of the paper is the ensemble technique for modeling wave runup rather than the erosion model calibration, however I found the sections focused on the transport coefficient, Cs, to break up the flow of the manuscript. Specifically, the section describing previous research on Cs (Lines 415 – 440) can be shortened and put into an appendix or supplemental information so the reader can focus instead on the results. I felt the most important information in this section was what Cs value was being used, how much it ranged, and specifically Lines 430 – 440."

Author Response to Comment 5: We acknowledge the Reviewer's request to shorten old Lines 415 – 440 (new Lines 420 Âň– 445). However, we feel that the information

regarding Cs is important for contextualizing the study and would prefer to leave it in the main body of the text. We would be happy to move this information to an Appendix but will leave the decision to the discretion of the Editor.

Comment 6: "Additionally, the results that appear in the discussion section (section 5.2) could be moved to the end of the results section. While these results are important for weighing the importance of Cs, they felt out of place in the discussion section."

Author Response to Comment 6: We agree with the Reviewer that moving Section 5.2 to the results section would improve the flow of the manuscript. As such, Sect. 5.2 in the original manuscript has now been moved to new Sect. 4.3 in the revised manuscript.

Comment 7: "Finally, this manuscript left me wondering what type of applicability this has to other locations. For example, is this method limited to locations with data?"

Author Response to Comment 7: This is a great question from the Reviewer and it is the topic of future work to determine how generalizable the runup predictor is and if site-specific data is required to apply the predictor to other locations, as now stated in Lines 637 – 638. "Future work is focused on using more data and additional inputs, such as offshore bar morphology and wave spectra, to improve the GP runup predictor developed here, testing it at different locations and integrating it into a real-time coastal erosion forecasting system."

Comment 8: "Lines 294 – 295: The authors may want to consider citing the technical methods for extracting wave runup as this is not a trivial task."

Author Response to Comment 8: Wave runup was extracted using a neural network runup tracking tool developed at Narrabeen Beach and which is available on GitHub. We are in the process of preparing a citable DOI for this package (Simmons et al. (2019)) and will include it in the final version of this manuscript during the proofing stage, before publication, on Lines 296–297: "Individual wave runup elevation on the

beach profile was extracted on a wave-by-wave basis from the LIDAR dataset (Fig. 2C) using the neural network runup detection tool developed by Simmons et al. (2019)."

Comment 9: "Lines 392 – 393: What is the resolution of SWAN model? 10m?"

Author Response to Comment 9: The SWAN model is based on a 10 m resolution grid. This has now been clarified in the manuscript at Lines 305 – 306: "...using a pre-calculated look-up table generated with the SWAN spectral wave model based on a 10 m resolution grid..."

Comment 10: "Figure 3: Why is the wave runup histogram plotted in red?"

Author Response to Comment 10: The wave runup histogram was originally plotted in red to identify it as the response variable in the GP mode. However, Fig. 3 has now been modified so that the wave runup histogram matches the color of the input variables.

Comment 11: "Does the blue display the runup prediction or the total water level prediction? It looks to me like there are tides within the blue values. Also, is it possible to plot some of the wave runup data on 7a? Authors show observations of erosion and dune toe, however, their model is predicting wave runup so that would be interesting to see in the time series sense too."

Author Response to Comment 11: The Reviewer is correct that the blue on Fig. 7 displays the total water level predictions (i.e., runup + water level). This is to illustrate how the dune erodes as the water level exceeds the dune toe in the L04 model. However, we agree with the Reviewer that it would be useful to see an example of just the wave runup prediction of the GP. Fig. 7 has now been modified to include this information.

Comment 12: "Line 35: The authors use the term 'significant wave period' to describe the wave period variable in runup formulations and then later mostly use 'peak wave period'. As 'significant wave period' is not used as typically, I'd recommend sticking with 'peak wave period' or simply 'wave period'."

Author Response to Comment 12: This is a good pick-up by the Reviewer, the term "significant wave period" has been replaced by "Peak wave period" which is used throughout the rest of the manuscript.

Comment 13: "Line 285: Please define SSE acronym, and if not used again, no acronym is necessary."

Author Response to Comment 13: This acronym has been replaced by south-southeast as it is not used again in the manuscript.

Comment 14: "Lines 446 – 447, Lines 509-510, Lines 545-546, Lines 595 – 596, Lines 619-620: repetition of similar variations of the following text, "drawing 10,000 samples takes than one second on a standard desktop computer." I'd recommend the authors say this a few times, then perhaps the term "computationally efficient" as this felt repetitive to read 5 times."

Author Response to Comment 14: This statement has now been removed from old Lines 545–546, 595–596, 619–620.

Please also note the supplement to this comment:
https://www.nat-hazards-earth-syst-sci-discuss.net/nhess-2019-81/nhess-2019-81-AC1-supplement.pdf

---

## Author Comment (AC2) · 5 Aug 2019

Please see the attached .pdf file for a formatted version of the author response document.

We thank both Reviewers for their time and effort in providing constructive feedback on our manuscript. Their comments have led to a much-improved manuscript with greater clarity in the description and purpose of the work presented. Below is our point-by-point response to the comments made and details of where the related changes have been made in the revised manuscript, in which they have also been highlighted. For clarity, Reviewer comments have been separated into key points which are addressed individually.

[Figure]

REVIEWER 2

Comment 1: "The authors make the bold (and most likely correct) statement that the development of a perfect deterministic parameterization of wave runup using only the typical inputs of beach slope, wave height, and wave period is improbable. They then go on to develop a GP runup model that has higher skill than the most typical deterministic runup model used today (Stockdon et al., 2006). However, to build this new model they still use the same three easily obtainable inputs. While perfectly reasonable for this paper's demonstration purposes, I am left wondering whether or not GP could be used to build an even better runup model if other input forcing dimensions were included? Figure 4 appears to have some structure in it, with low values of R2 overpredicted and high values underpredicted. Can we learn something from this? Even a few suggestions and/or speculations from the authors would be welcome about machine learning directions for developing even better runup models."

Author Response to Comment 1: We expect that the performance of the runup predictor could potentially be improved using additional inputs in the future. We feel that useful inputs to include in the next iteration of the runup model would be bar morphology (i.e., presence/absence of an offshore bar) and wave spectra. Unfortunately, this data is not as easily available as Hs, Tp and beach slope, but deserves to be collected. This speculation and direction for future work has been included in Lines 637 – 638: "Future work is focused on using more data and additional inputs, such as offshore bar morphology and wave spectra, to improve the GP runup predictor developed here, testing it at different locations and integrating it into a real-time coastal erosion forecasting system."

Comment 2: "In developing the input Hs and Tp time series for both the creation of the runup model and for the ultimate test against the dune erosion event, it is mentioned that SWAN is used to transform all conditions into the nearshore before being linear back shoaled. Did the authors really run 100s to 1000s of individual SWAN simulations? This effort seems like it must have had a high computational cost? Since the

paper emphasizes the efficiency of the GP runup model some more detail of this step in the process is warranted. Have the authors considered developing simple look up tables, or better yet, a GP model of SWAN to simplify this stage of the process?"

Author Response to Comment 2: The Reviewer makes the good point that running 1000s of SWAN simulations to calculate nearshore wave conditions would be impractical. Here, as the Reviewer suggests, we used a pre-calculated look-up table for Narrabeen Beach to transform the offshore wave conditions, which is computationally cheap. This has now been clarified at Lines 304 – 306: "...offshore wave data were first transformed to a nearshore equivalent (10 m water depth) using a pre-calculated look-up table generated with the SWAN spectral wave model based on a 10 m resolution grid..."

Comment 3: "The decision to use MDA for developing the training data seems sound. However, a list, or discussion of other possible space filling algorithms might be useful for readers embarking on their own GP applications."

Author Response to Comment 3: A short discussion on alternative data selection methodologies has now been added at Lines 264–267: "While alternative data-splitting routines are available, including simple random sampling, stratified random sampling, self-organizing maps and k-means clustering (Camus et al., 2011), the MDA routine used in this study was found in preliminary testing (not presented) to produce the best GP performance with the least computational expense."

Comment 4: "I commend the authors for their relatively parsimonious and clear explanation of GP theory in section 2.1. However, I suspect that this treatment will still be an occasionally opaque to some readers (including this reviewer). My only suggestion here is to continue to work on describing machine learning approaches such as GP in as clear of terms as possible. This paper does this as well as I have seen."

Author Response to Comment 4: We certainly agree with the Reviewer that the clear communication of machine learning methods in general is critical to the proper implementation and interpretation of these methods to coastal problems. We very much appreciate the Reviewer's positive feedback on our attempt to do that in this manuscript.

Comment 5: "Line 81-82: Maybe add to this growing body of literature by including: Parker, K., P. Ruggiero, K. Serafin, and D. Hill. 2019. "Emulation as an Approach for Rapid Estuarine Modeling." Coastal Engineering 150: 79–93. https://doi.org/10.1016/j.coastaleng.2019.03.004."

Author Response to Comment 5: We thank the Reviewer for providing this excellent paper which became available during the review process of the current paper and which we have read with interest. It very nicely supports the (presently limited) body of literature around Gaussian processes in coastal applications and we have now cited it in the text at Line 82: "Recent work has specifically used Gaussian processes to model coastal processes such as large scale coastline erosion (Kupilik et al., 2018) and estuarine hydrodynamics (Parker et al., 2019)."

Comment 6: "Line 235-236: I thank the author for identifying which toolkit they used in developing the runup GP model. However, it might be helpful for a broad group of readers if the authors listed other potential toolkits that could also have been used – say for example in Matlab, or R?"

Author Response to Comment 6: A comment on alternative languages/software for developing Gaussian Processes has now been added at Lines 234Âř–236: "For the Reader unfamiliar with the Python programming language, alternative programs for developing Gaussian Processes include Matlab (Rasmussen and Nickisch, 2010) and R (Dancik and Dorman, 2008; MacDonald et al., 2015)."

Comment 7: "Line 544-546: The statement about 10,000 samples taking less than one second on a standard desktop computer is repetitive at this point."

Author Response to Comment 7: This statement has now been removed from old Lines 544–546.

Please also note the supplement to this comment:
https://www.nat-hazards-earth-syst-sci-discuss.net/nhess-2019-81/nhess-2019-81-AC2-supplement.pdf